# Estimating the time of Highly Pathogenic Avian Influenza virus introduction into United States poultry flocks during the 2022/24 epizootic

Amos Ssematimba[1,2]*, Sasidhar Malladi[3], Peter J. Bonney[1], Kaitlyn M. St. Charles[1], Holden C. Hutchinson[3], Melissa Schoenbaum[3], Rosemary Marusak[1], Marie R. Culhane[1], Carol J. Cardona[1]

1 Secure Food Systems Team, University of Minnesota, Saint Paul, Minnesota, United States of America, 2 Department of Mathematics, Faculty of Science, Gulu University, Gulu, Uganda, 3 U.S. Department of Agriculture, Animal and Plant Health Inspection Service, Veterinary Services, Center for Epidemiology and Animal Health, Fort Collins, Colorado, United States of America

* amos.ssematimba@gmail.com

**Data Availability Statement:** The data used as part of this study were collected by USDA as part of

## Abstract

Following confirmation of the first case of the ongoing U.S. HPAI H5N1 epizootic in commercial poultry on February 8, 2022, the virus has continued to devastate the U.S. poultry sector and the pathogen has since managed to cross over to livestock and a few human cases have also been reported. Efficient outbreak management benefits greatly from timely detection and proper identification of the pathways of virus introduction and spread. In this study, we used changes in mortality rates as a proxy for HPAI incidence in a layer, broiler and turkey flock together with diagnostic test results to infer within-flock HPAI transmission dynamics. Mathematical modeling techniques, specifically the Approximate Bayesian Computation algorithm in conjunction with a stochastic within-flock HPAI transmission model were used in the analysis. The time window of HPAI virus introduction into the flock (TOI) and the adequate contact rate (ACR) were estimated. Then, using the estimated TOI together with the day when the first HPAI positive sample was collected from the flock, we calculated the most likely time to first positive sample (MTFPS) which reflects the time to HPAI detection. The estimated joint (i.e., all species combined) median of the MTFPS for different flocks was six days, the joint median most likely ACR was 6.8 newly infected birds per infectious bird per day, the joint median $R_0$ was 13 and the joint median number of test days per flock was two. These results were also grouped by species and by epidemic phase and discussed accordingly. We conclude that this findings from this and other related studies are beneficial for the different stakeholders in outbreak management. We recommend that combining TOI analysis with complementary approaches such as phylogenetic analyses is critically important for improved understanding of disease transmission pathways. The estimated parameters can also be used to parametrize mathematical models that can guide the design of surveillance protocols, risk analyses of HPAI spread, and emergency preparedness for HPAI outbreaks.

disease outbreak response activity purposes and are considered confidential business information with legal repercussions in case of unauthorized sharing of the same. Anonymized data may be made available upon request, for statistical purposes only and such requests to access the datasets should be directed to HCH at USDA via the email: vs.ceah@usda.gov.

**Funding:** This study was funded through a cooperative agreement between the Center for Epidemiology and Animal Health (CEAH) of the USDA, Animal and Plant Health Inspection Service (APHIS) Veterinary Services (VS) and the University of Minnesota (UMN) as USDA Award # AP22VSSP0000C051 (Quantitative Analysis and Support for Agricultural Animal Disease Emergency Preparedness and Response in the U.S.). The findings and conclusions in this document are those of the authors and should not be construed to represent any official USDA or U.S. Government determination or policy. The funders had no role in study design, data collection and analysis, decision to publish, or preparation of the manuscript.

**Competing interests:** The authors have declared that no competing interests exist.

## Introduction

The 2022/24 H5N1 Highly Pathogenic Avian Influenza (HPAI) clade 2.3.4.4b outbreak in North American poultry and other species was preceded by detections in wild birds. The initial detection was from a deceased Great Black-backed Gull submitted to the Canadian Wildlife Health Cooperative in late 2021 [1]; subsequently, the first detections in the United States occurred in January 2022 and were among wild ducks sampled for USDA active wild bird surveillance [2]. The outbreak was caused by a virus closely related to the 2015 outbreak virus [3], but with some unique transmission characteristics (e.g., lateral spread vs. independent introductions via wild birds) [4].

For the ongoing HPAI outbreak in the U.S., the first case in commercial poultry was confirmed on February 8, 2022, on a turkey farm in Dubois County, Indiana. Since then, detections have been reported in other commercial poultry production types including broilers, table egg pullets and layers, ducks, raised-for-release upland game birds and in live bird markets [4]. The virus continues to spread to poultry and other food animal sectors in the U.S. As of June 4, 2024, a total of 1,149 flocks, of which 495 are commercial poultry, have been confirmed in 48 states, affecting 96.8 million birds due to identified and controlled outbreaks [5]. Subsequently, the expansion of the outbreak has resulted in over one billion U.S. dollars in federal expenditures for control efforts and indemnity payments (accounting for > $715 million as of February 21, 2024).

As of June 4, 2024, the United States has detected HPAI H5 virus in a total of 259 wild mammals; these mammals include, but are not limited to, cats, mice, foxes, skunks, mountain lions, harper seals, bobcats, and bears [4, 6]. Of note, HPAI H5N1 was for the first time reported in domestic ruminants in March 2024 with infections being reported in kid goats on a farm where a poultry flock had tested positive for the same virus. In the same month, infections were also reported for the first time in dairy cows [7]. Additionally, 14 human cases have been associated with the ongoing poultry and dairy cattle outbreaks; these cases have occurred among individuals involved with infected poultry flock depopulation and workers on infected dairy farms. Person-to-person spread has not been reported and the current public health risk is deemed to be low (CDC, 2024b).

As part of outbreak investigation, full genome sequencing results indicated that independent wild bird introductions were the primary mechanism of HPAI virus introduction onto poultry premises in this outbreak [8] and there have been reports of virus spread between cows and poultry [9]. In contrast, the 2014/15 U.S. HPAI H5N2 and H5N8 outbreak was primarily driven by lateral transmission of virus between farms [10]. The differences in spread mechanisms, as well as the larger geographic scope of the 2022/2024 outbreak compared to the 2014/15 outbreak, necessitated further examination into risk factors for pathogen introduction and biosecurity practices. To elucidate the possible discrepancies between the two outbreaks, the USDA performed case-control studies for commercial turkey [11] and commercial table egg layer flocks [12]. Collectively from the two studies, the identified risk factors for HPAIV introduction included being in a disease control zone, increased sightings of wild waterfowl in the vicinity of the poultry premises, and offsite rendering of farm byproducts, among others.

Efficient outbreak management benefits greatly from timely detection and proper identification of the pathways of virus introduction and spread. For HPAI in poultry, surveillance for clinical signs such as any unexplained illness or increase in mortality, and decreased egg production [13] as well as decreased feed and water consumption [14] is critically important. Using changes in mortality rates, egg production, and water and feed consumption as proxies for HPAI incidence in a flock, mathematical modeling techniques can be utilized to estimate

within-flock disease transmission parameters and subsequently estimate the time window of pathogen introduction into the flock.

For example, for low pathogenicity avian influenza, Gonzales et al. [15] used egg production data while Pinsent et al. [16] and Bonney et al. (2021) used diagnostic testing data to estimate the transmission rate within a flock and time of virus introduction (TOI). For HPAI, Bos et al. [17], Hobbelen et al. [18], Vergne et al. [19] and Hayama et al. [20] all used mortality data to estimate the time of HPAI introduction into a flock and Ssematimba et al. [21] used mortality data to estimate the adequate contact rate (ACR). In all, outcomes for such an analysis can inform optimization of tracing efforts and epidemiological investigations. Narrowing down the time window for possible virus introduction helps to identify the potential routes of virus introduction and enhances the understanding of the pattern of disease spread.

Here, we use stochastic disease transmission mathematical models in conjunction with an Approximate Bayesian Computation (ABC) algorithm, parametrized by data from the most recent literature as well as field diagnostic test results and daily mortality data from the 2022/24 U.S. H5N1 HPAI outbreak, to estimate the TOI and ACR for the virus. The estimated TOI is then used to determine the most likely median time between flock exposure to the virus and collection of the first rRT-PCR positive sample (MTFPS). The potential effects of bird species and epidemic phase on the resulting estimates were investigated as well as the difference in number of test days across species and epidemic phase.

## Materials and methods

### Data

The data used in this analysis included daily mortality records and diagnostic laboratory submission reports and test results. Mortality records are provided to the USDA for indemnity purposes; laboratory documents for positive premises are recorded in USDA's Emergency Management Response System. This data was deidentified and provided to the Secure Food Systems (SFS) Team at the University of Minnesota for analysis via a material transfer agreement.

Flock data provided to the SFS team reflect a convenience sample of infected premises. Infected premises which were the first detection in a State and/or thought to be involved in cases of lateral transmission were selected for analysis to support ongoing field investigations. Furthermore, premises with inadequate data (e.g., mortality records provided in aggregate instead of daily, unknown sampling schemes) or that had abnormal baseline mortality patterns, likely due to underlying comorbidities (e.g., avian metapneumovirus), were excluded. Commercial premises infected between February 2022 and December 31, 2023 were eligible for inclusion. At this time, 452 commercial premises infected with HPAIV were confirmed detected. Detections included 39 commercial broiler premises, 55 commercial table egg layer premises, and 308 commercial turkey premises. The SFS team also directly received some data from affected companies after signing confidentiality agreements and these were also included in the analysis.

### Disease transmission model

A stochastic SEIR (i.e., susceptible, latently infected, infectious, and removed (dead)) individual bird-based HPAI transmission model was used to simulate the disease induced mortality and infection prevalence. The model assumed that a flock comprised of a closed homogeneously mixing population with infection introduced by a single latently infected bird. The model followed the approach described in [22, 23] and simulated individual bird transitions between different disease states over time post exposure. The number of birds that transitioned

**Table 1. Input prior distribution parameters used in the ABC approach to estimate the adequate contact rate and time of virus introduction.**

| Parameter Name | Description | Distribution | |
|---|---|---|---|
| | | **Turkeys** | **Layers/Broilers** |
| Adequate Contact Rate (ACR) | Daily average number of contacts a bird has with other birds that are sufficient to transmit infection | Uniform (min = 0.2, max = 20) per day | Uniform (min = 0.2, max = 20) per day |
| Latent Period Length Distribution | Length of the interval when a bird is latently infected and is not infectious | Gamma (shape = 4.38, scale = 0.16); mean = 0.72 days | Gamma (shape = 2.54, scale = 0.33); mean = 0.84 days |
| Mean infectious period | Prior distribution for the mean infectious period | Uniform (1.93–4.0) days | Uniform (0.74–4 days) |
| Shape parameter for infectious period | Prior distribution for shape parameter of gamma distributed infectious period | Uniform (1–20) | Uniform (1–20) |

from the susceptible to the latently infected state in a simulation time step (set at 0.01 days) was simulated from a binomial distribution where the probability of infection depended on the contact rate and prevalence of infectious birds in the barn. Individual bird transitions from the latently infected to the infectious state and from the infectious to the removed (i.e., dead or recovered) state were simulated using gamma distributions. The stochastic simulation model output was used in conjunction with an ABC algorithm to estimate the transmission parameters and time of HPAI introduction into the flock.

The prior distributions for the bird-level latent and infectious period used in the model (Table 1) were obtained using Markov Chain Monte Carlo (MCMC) algorithms on data from challenge studies in turkeys and chickens with a current outbreak isolate [3]. Relatively wide priors were used for the ACR given the variability in estimates from outbreak data and experimental studies and the need to consider the potential impact of virus strain and management practices [21, 24, 25].

## Approximate Bayesian Computation (ABC) algorithm

An ABC algorithm in conjunction with a within-flock HPAI disease transmission model were used to estimate the TOI, ACR, and other parameters based on the daily mortality and diagnostic test results. Water consumption data was available and used only for the reported index commercial poultry case in the U.S. A goodness of fit metric $\psi$ was used to compute the distance between the disease transmission model predictions and observed data as shown in Eq 1.

$$\psi = \frac{1}{L} \sum_i \left( \frac{M_{sim,i} - M_{obs,i}}{\sigma_m} \right)^2 + \kappa \tag{1}$$

Here $M_{sim,i}$ and $M_{obs,i}$ are respectively the simulated and observed daily mortality on day $i$, $\sigma_m$ is the standard deviation of the baseline mortality estimated from the data and $\kappa$ is the testing cost. The parameter $L$ is the number of sequential days of daily mortality that were included in the fitting procedure (set to 7 days for most flocks). Note that a large value of $L$ can result into overfitting the model to normal mortality. Similar distance measures based on squared difference between the observed and model predicted mortality have been utilized in other outbreak analyses [26, 27].

Given the importance of the diagnostic test results, the testing cost $\kappa$ is set to a large value relative to the daily mortality cost component (i.e., 1000) if the observed and simulated diagnostic test results do not match and set to zero otherwise. Given this testing cost value, only iterations in which the simulated test results match with the observed results are accepted in the ABC algorithm. The matching of observed and simulated test results in the acceptance criteria of the ABC algorithms was also prioritized in Nezworski et al. [27]. The distance between

simulated and observed water consumption, when available, was computed similarly to the approach used for mortality.

An ABC-MCMC algorithm based on Marjoram et al. [28] with modifications detailed in [29, 30] was used. Specifically, the algorithm includes a calibration step for $n$ simulations with a tolerance level $\varepsilon$ and threshold distance $\psi_e$ such that for a disease transmission model simulation with randomly selected parameter values from the priors, the probability that the distance is less than $\psi_e$ i.e., $P(\psi \leq \psi_e)$ is $\varepsilon$. The standard deviation of the parameter values in the accepted iterations from the calibration step also informed the width of the proposal distribution. In addition, the selected iterations in the calibration step were used as starting points for independent MCMC chains run in parallel among many Central Processing Units, similar to the approach described in [29].

Convergence of the ABC-MCMC algorithm was evaluated via trace plots, the Gelman-Rubin potential scale reduction factor, Heidelberger and Welch's convergence diagnostic and other diagnostics provided in the R package *coda* [31]. In practice, the above convergence diagnostics were acceptable when running a million or more iterations with minimal difference in posterior distribution means among different simulations. For each poultry house evaluated, the ABC-MCMC algorithm was run for 250,000 calibration iterations and one million MCMC iterations with a tolerance level of 0.0001.

## Analysis

The current analysis focused on outbreaks in commercial broilers, egg layers, and turkeys, given that those three species are the most dominant species in the U.S. poultry industry and accounted for majority of the reported outbreaks. First, as an example of how the model-derived data fits to the observed field data, we shall present a graph (Fig 1) on mortality data for the index commercial case.

In order to understand how species-type and epidemic phase may impact the study outcomes, premises were grouped by species (i.e., broiler, layer and turkey) and by epidemic phase (i.e., end- and post- July'22). Then, both joint and species- and phase- specific outputs were generated. Note that there were no new reported HPAI cases in commercial broilers,

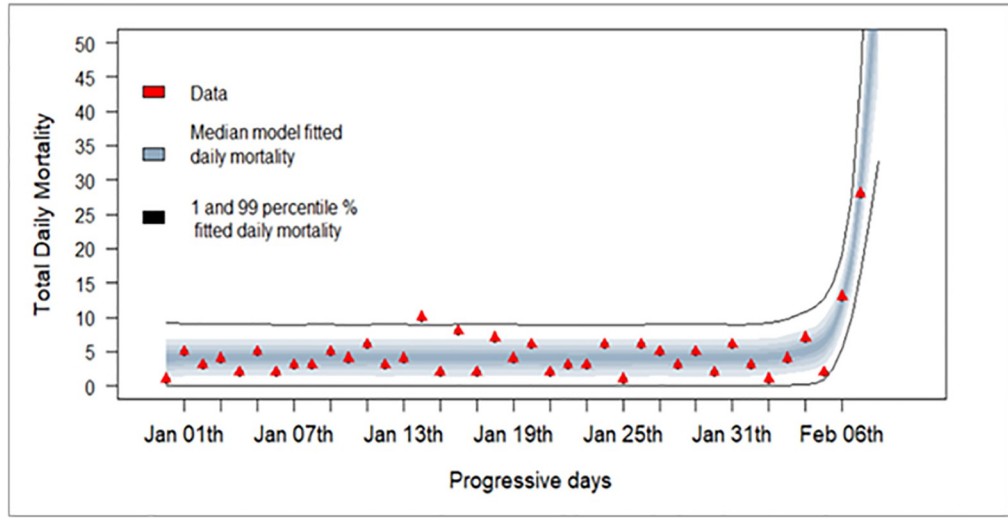

**Fig 1. Observed (red triangles) and model predicted daily mortality (grey band) for the index commercial poultry case of the 2022/24 U.S. H5N1 HPAI epidemic.**

layers, and turkeys from July 26 to August 22, 2022 [5]. Therefore, in this analysis the epidemic phase cutoff was set to July 31, 2022 and of the then 402 reported outbreaks in commercial broilers, egg layers, and turkeys, 174 outbreaks occurred prior to July 26, 2022 and 228 occurred between August 22, 2022 and December 31, 2023.

All the quantitative data (model- or field- derived) was tested for normality using the Shapiro-Wilk normality test [32] and based on the outcomes of this test (i.e., non-normality), output distributions were summarized using the median and the lower and upper quartiles. Where necessary, statistical comparisons between the estimated medians were performed using the Wilcoxon rank-sum test [33] and in all statistical tests, a significance level of $p < 0.05$ was used.

Outputs of interest include the MTFPS, the within-flock ACR and $R_0$. Given the fair amount of variation in ACR between flocks, we also categorized flocks based on the estimated median ACR into four classes namely, slow (ACR $< 1$), moderate ($1 \leq$ ACR $< 3$), fast ($3 \leq$ ACR $< 7$), and very fast (ACR $\geq 7$). We also summarize the number of test days (obtained from the available testing data of a given flock during the critical exposure period/ time window). Due to potential overfitting to variations in normal mortality, flocks with no spike in mortality (i.e., where mortality does not exceed the predetermined species-specific mortality trigger threshold) by the time of detection were assumed to potentially underestimate the ACR and $R_0$ and were hence omitted from use in that part of the analysis. A sensitivity analysis was carried out to validate this assumption. All analyses were performed using version 4.3.3 of R software [34].

## Results

The subset of premises analyzed reflect 18% (72 out of 402) of all infected broiler (9 out of 39), layer (18 out of 55), and turkey (45 out of 308) premises during the data period; split by epidemic phase, 24% (42 out of 174) of premises during end -July'22 and 13% (30 out of 228) of premises post-July'22 were analyzed.

For the reported index commercial poultry case of the current outbreak in the U.S., the observed and model predicted outcomes for mortality data are presented in Fig 1. The estimated daily mortality from the model closely match the field mortality data for this flock, indicating a reasonable fit.

A summary of the number of test days and the results on the estimated MTFPS, within-flock ACR, and $R_0$ are presented in Table 2. Note that 15 (only two of which were after July

**Table 2. Estimated median, lower and upper quartiles for distributions of the number of test days, the most likely time to first positive sample (MTFPS), the within-flock adequate contact rate (ACR), and the basic reproduction number ($R_0$) for the U.S. 2022/24 H5N1 HPAI epidemic.**

| Species | Epidemic Phase | Flocks | Test days: median (LQ, UQ)[c] | MTFPS[a] (days): median (LQ, UQ)[c] | ACR[b] (per day): median (LQ, UQ)[c] | $R_0$: median (LQ, UQ) |
|---|---|---|---|---|---|---|
| All species | Entire | 72 | 2.0 (1.0, 3.0) | 6.0 (4.0, 7.0) | 6.8 (3.0, 17.6) | 13.0 (7.0, 29.0) |
| | End-July'22 | 42 | 2.0 (1.0, 2.0) | 5.0 (5.0, 7.0) | 6.8 (3.0, 16.9) | 15.0 (6.0, 29.0) |
| | Post-July'22 | 30 | 2.0 (1.0, 4.0) | 6.0 (5.0, 7.0) | 6.6 (3.3, 17.7) | 13.0 (7.8, 31.0) |
| Broiler | Entire | 9 | 2.0 (1.0, 2.0) | 7.0 (6.0, 9.0) | 5.0 (1.8, 5.3) | 6.0 (4.0, 9.0) |
| Layers | Entire | 18 | 3.0 (1.3, 4.8) | 4.0 (3.0, 5.8) | 9.1 (2.9, 17.4) | 13.5 (5.3, 25.3) |
| Turkey | Entire | 45 | 1.0 (1.0, 2.0) | 6.0 (5.0, 7.0) | 9.4 (3.3, 18.4) | 25.5 (9.3, 38.8) |

[a]MTFPS is the number of days between the estimated most likely time of virus introduction and the day when the first HPAI virus positive sample was collected.

[b]ACR is the adequate contact rate and defines the number of new infections caused by a typical infectious bird per day.

[c]LQ and UQ are the lower (25th percentile) and upper (75th percentile) quartiles of the distribution of the output.

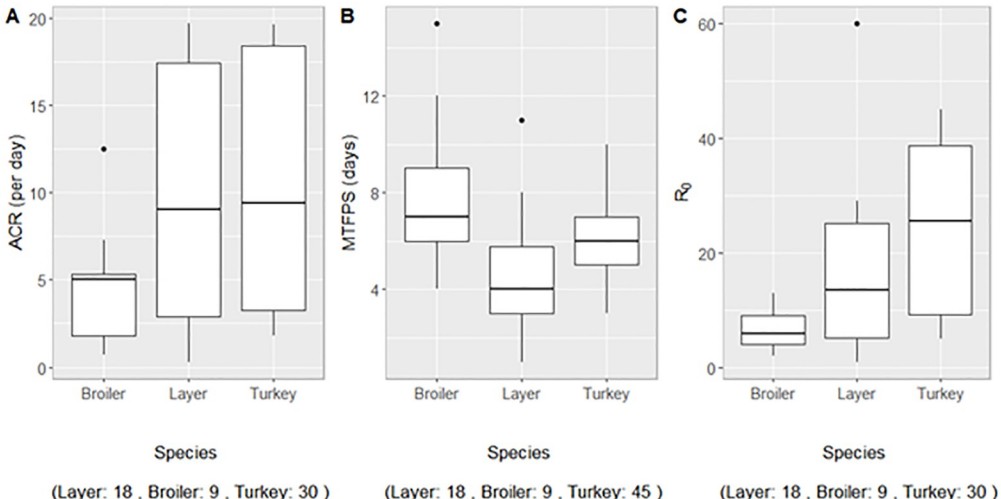

**Fig 2. Estimated species-specific within-flock adequate contact rate (ACR), most likely time to first positive sample (MTFPS), and basic reproduction number ($R_0$) during the U.S. 2022/24 H5N1 HPAI outbreak.**

2022) of the turkey flocks did not show a substantial spike in mortality by the time of their detection and were hence only used in the estimation of MTFPS and the number of test days. For all infected flocks analyzed as of December 31, 2023, the estimated joint (i.e., all species combined) median of the MTFPS for different flocks was six days, the joint median most likely ACR was 6.8 newly infected birds per infectious bird per day, the joint median $R_0$ was 13 and the joint median number of test days per flock was two (Table 2).

When grouped by species (Table 2 and Fig 2), the median number of test days per flock was lowest in turkeys and highest in layers with one and three days respectively. The difference was significant between turkeys and layers ($p = 0.01$), insignificant between broilers and layers with upper quartiles of two and 4.8 test days respectively ($p = 0.08$), and insignificant between broilers and turkeys. The median MTFPS was longest in broilers at seven days and shortest in layers at four days and all differences in median MTFPS across species were significant, specifically when comparing broilers vs. layers ($p = 0.004$), broilers vs. turkeys ($p = 0.0005$), and turkeys vs. layers ($p = 0.011$).

The median ACR was lowest in broilers at five newly infected birds per infectious broiler bird per day and highest in turkeys at 9.4 birds per day. The difference in median ACR was significant between broilers and turkeys ($p = 0.02$), insignificant between broilers and layers ($p = 0.19$) and insignificant between layers and turkeys. Lastly, at 25.5 new infections per infectious bird, the median $R_0$ was highest in turkeys and lowest in broilers at six. The difference in median $R_0$ was significant between broilers and turkeys ($p = 0.0005$) and between layers and turkeys ($p = 0.03$), and was insignificant between broilers and layers ($p = 0.099$).

When grouped by epidemic phase (Table 2 and Fig 3), the median MTFPS during the first phase was five days compared to six days in the second phase, the median $R_0$ was slightly lower in the first phase and the median ACR varied only slightly. However, between the two phases, the differences in joint median values of MTFPS, ACR and $R_0$ were all insignificant and the difference in the number of test days was also insignificant ($p = 0.19$), a result that is also echoed in the upper quartile which increased from two to four test days from the first to the second phase.

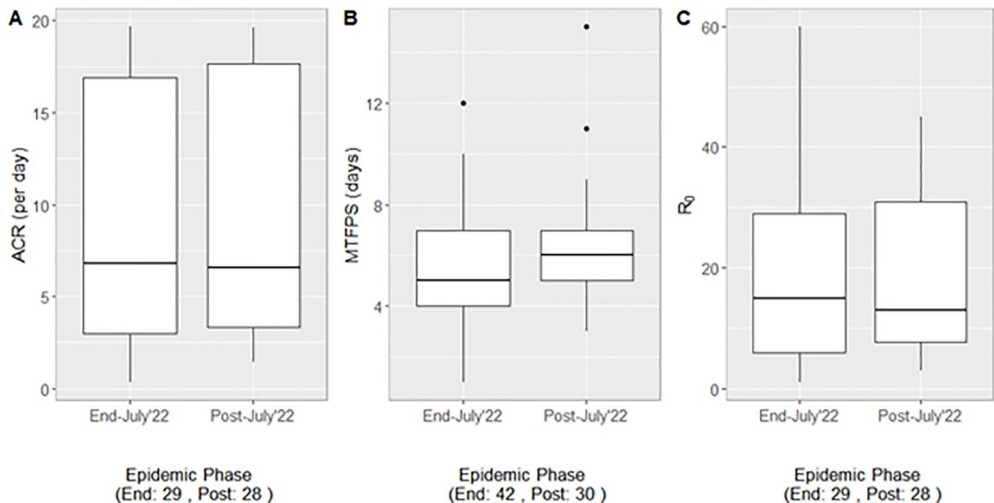

**Fig 3. Estimated within-flock adequate contact rate (ACR), most likely time to first positive sample (MTFPS), and basic reproduction number ($R_0$) for the two phases of the U.S. 2022/24 H5N1 HPAI outbreak.**

When grouped by both epidemic phase and species, the difference in median MTFPS across the two phases was insignificant within broilers and turkeys. Of note, although statistically insignificant ($p = 0.12$), the median MTFPS among layers increased from 4 to 5.5 days and the upper quartile increased from 4.3 days to six days between the two phases. The median (and quartiles) of the number of test days for the first phase was 2.5 (1.8, 4.8) for layers, 2.0 (1.5, 2.0) for broilers, and 1.0 (1.0, 2.0) for turkeys and in the second phase it was 3.5 (1.5, 4.8) for layers, 1.5 (1.0, 2.0) for broilers, and 2.0 (1.0, 3.8) for turkeys. The difference in median number of test days across phases was only significant within turkeys ($p = 0.04$).

When categorized by ACR, four flocks (i.e., one broiler and three layer) had a slow spread, 10 flocks (i.e., 5 turkey, 3 broiler and 2 layer) had a moderate spread, 15 flocks (i.e., 8 turkey, 3 broiler and 4 layer) had a fast spread, and 28 flocks (17 turkey, 2 broiler and 9 layer) had a very fast spread. Lastly, for the 15 turkey flocks that did not exhibit a spike in mortality, their estimated median ACR was 1.7 newly infected birds per infectious bird per day and the corresponding median $R_0$ was six birds.

## Discussion

In this study, we estimated the time window for HPAI introduction and within-flock disease transmission parameters for 72 commercial poultry flocks that became infected during the U. S. 2022/24 H5N1 HPAI outbreak using diagnostic test results and mortality data. Combining TOI analysis with complementary approaches such as phylogenetic analyses and field epidemiological investigations is critically important for improved understanding of disease transmission pathways. When all flocks were grouped by phase without consideration for species, there was only a slight decrease from a median ACR of 6.8 to 6.6 birds per day and in $R_0$ from 15 to 13 birds when observing differences between the first phase, End-July'22, and the second phase, Post-July'22. However, the observed decreased between phases was not significant. Also, when grouped by both species and epidemic phase there were no significant differences observed. The negligible change in these estimates based on phase may be indicative of the fact that the virus did not change much in its transmissibility characteristics across the two phases.

By species alone, the estimated median ACR and $R_0$ were highest in turkey flocks and lowest in broiler flocks. Moreover, turkey flocks were observed to have significantly higher $R_0$ in comparison to broiler flocks. The differences in estimated median ACR and $R_0$ indicate that the virus was significantly more transmissible in turkeys than in broilers and layers (Table 2). Higher transmissibility of HPAI virus between turkeys in comparison to between chickens has also been reported in previous studies [35, 36]. Additionally, findings in the present study are in agreement with the outcomes from experimental studies by Pantin-Jackwood et al. [3] in which the authors demonstrated that successful infection of chickens required a higher inoculation dose compared to turkeys, and chickens had shorter mean death times than turkeys. Similar observations were made in the study by Puranik et al. [36] wherein the observed extended mean time to death in turkeys results in a longer virus shedding period and hence enhances transmission to contact birds (i.e., higher $R_0$).

However, comparing findings from studies involving different virus strains should be done with caution. Note that HPAI virus strain-specific characteristics [37], production type, and flock management practices may influence within flock disease spread dynamics. As examples of closely related studies using field HPAI outbreak mortality data in commercial turkey flocks to estimate disease parameters, an ACR of 3.2 (95% CI: 2.3, 4.3) per day and $R_0$ of 12.8 (95% CI: 9.2, 17.2) were estimated for the 2014/15 U.S. H5N2 epidemic [21] and, an ACR of 3.37 (95% CI: 0.97, 11.74) per day was estimated for the Dutch 2003 H7N7 epidemic [24]. Generally, when comparing the two large U.S. HPAI epidemics, the 2022/24 outbreak virus strain was more infectious than that of 2014/15 outbreak virus strain [3].

Overall, these results suggest that the most likely time to first positive sample and detection was relatively long for commercial broiler flocks (7.0 [6.0, 9.0] days) compared to other species and the number of test days were few (2.0 [1.0, 2.0] days). The longer time to detect paired with a smaller number of test days is an indicator of a likely negative correlation between the two measures. In contrast, the results were the exact opposite in egg layers (MTFPS: 4.0 [3.0, 5.8] days; Median test days: 3.0 [1.3, 4.8] days). Possible explanations for this apparently "unbalanced" delay in HPAI detection and frequency of diagnostic testing may partly lie in each species' reasons for testing. Reasons for testing range from a farm's participation in supporting continuity of business initiatives to farm staff's ability to observe clinical signs and to the likelihood of a farm falling within a disease control area and surveillance zone.

For example, layer premises that fall within a disease control area must perform regular diagnostic testing to meet surveillance requirements imposed by the state animal health regulatory agency as well as part of obtaining egg movement permits. The diagnostic testing related to egg movements typically occurs at higher frequency than the testing required for single-use permits related to terminal or transfer live bird movements that are common in turkey or broiler production. The more frequent testing required for egg laying flocks that enables egg movements off farms may explain the higher number of test days, and hence the relatively shorter MTFPS, in layers compared to other analyzed species that are focused on live bird movements.

We hypothesize that the longer MTFPS observed in broilers may be influenced by the lower ACR observed and/or by sampling requirements and production cycles. All poultry premises are not required to sample and test their birds under control area surveillance until birds are a minimum of 21 days of age. Given that the average market age of commercial broilers is 47 days [38], the window for required testing is smaller for broiler flocks compared to market ages of grow-out turkeys (i.e., 98 to 140 days) [39]. Additionally, the majority of the broilers in the U.S. are raised in the southeastern part of the country where there were many fewer HPAI cases and thus, many fewer control areas. Therefore, susceptible broiler farms that ended up

becoming infected may not have been as likely to end up in preexisting control areas, and were thus not as often subject to control area-mandated surveillance testing in the first place.

Note that over the duration of the entire outbreak, commercial turkey flocks were most frequently affected. Given the outbreak trend in the second phase, we hypothesize that more turkey premises ended up in control areas and surveillance zones leading to more testing. The increased testing of turkeys may also reflect changes in continuity of business-related testing due to (perhaps) seasonal holiday turkey market dynamics in the U.S. The observed changes in testing frequency may also reflect shift in mindset, shift in location of new outbreaks and/or seasonal dependence of testing, for example, in turkey flocks where producers in high-risk areas of fall wild bird migration tend to test more during that period. However, all these hypotheses require further analysis to elucidate them.

Results from the sensitivity analysis validated the study assumption that including flocks without a spike in daily mortality by the time of detection in the analysis would underestimate the ACR and consequently, $R_0$. Both measures of disease spread were much lower for the flat-mortality turkey flocks than those of the other turkey flocks analyzed. Notwithstanding, including without a spike in daily mortality by the time of detection flocks in the analysis did not affect the trend observed in the species-specific comparison of outcomes.

There are some possible limitations to approaches used in this study. For example, as revealed in the outcomes of the sensitivity analysis, more reliable estimates for disease spread rate parameters require that the within-flock disease spread reaches the exponential growth phase of daily mortality. The other likely limitation lies in the inability to delineate the upper bound of the credible interval of the ACR for flocks with very fast spread. In such flocks, this approach may lead to higher upper bounds that are heavily dependent on the set priors. This effect has also been reported previously in a study that estimated ACR for low pathogenicity avian influenza in the U.S. [23].

Estimating the time of HPAI virus introduction on individual farms provides valuable information for epidemiologic investigations and outbreak response and is beneficial for the different stakeholders in outbreak management. This has been demonstrated during the response to the two large HPAI outbreaks in the U.S. where the analyses played a key role in supporting rapid field tracing and epidemiological investigations to identify and rule out potential routes of exposure by facilitating the narrowing down of the time window of possible virus introduction for each flock. On the producers' end, knowing the estimated TOI on their own farms is beneficial for their individual understanding of; 1) how they became infected, 2) gaps that might exist in their biosecurity plans, and 3) knowing what risky practices they may be engaging in accidentally or intentionally. On the other hand, cumulative TOI results over the course of an outbreak benefit regulators, epidemiological investigators, academicians, and researchers by facilitating the understanding of how outbreaks are expanding, how viruses might be changing, and how outbreak response needs to adapt. Lastly, the estimated parameters can inform models used for surveillance design, risk analysis, and emergency preparedness.

The current study was mainly based on disease mortality trends, clinical signs and diagnostic test results. However, access to and implementation of other types of production data e.g., water and feed consumption or egg production data can reduce the uncertainty in model derived estimates. The approaches and outcomes of this study highlight the value of closely monitoring mortality to quickly identify disease issues in a flock. Because trends may vary across production type, a thorough understanding of species- and production- specific trends is important. For example, when comparing HPAI spread within a broiler and turkey flock, water and feeding systems may alter the disease spread dynamics since broiler commonly use drinker while turkeys use plassons.

## Acknowledgments

The authors acknowledge the contribution of all the other Secure Food Systems team members at the University of Minnesota namely, David Halvorson, Catherine Alexander, Margret Tavai-Tuisalo'o, Mickey Leonard, Miranda Medrano, Sylvia Wanzala, Timothy Goldsmith, and Cesar Corzo. We also thank Erica Spackman of SEPRL for the discussions and experimental data and the Minnesota Supercomputing Institute (MSI) at the University of Minnesota for providing resources that contributed to the research results reported in this paper.

## Author Contributions

**Conceptualization:** Amos Ssematimba, Sasidhar Malladi, Peter J. Bonney, Holden C. Hutchinson, Marie R. Culhane, Carol J. Cardona.

**Data curation:** Amos Ssematimba, Sasidhar Malladi, Peter J. Bonney, Holden C. Hutchinson, Melissa Schoenbaum, Rosemary Marusak.

**Formal analysis:** Amos Ssematimba, Sasidhar Malladi, Peter J. Bonney, Holden C. Hutchinson.

**Funding acquisition:** Marie R. Culhane, Carol J. Cardona.

**Investigation:** Amos Ssematimba, Sasidhar Malladi, Kaitlyn M. St. Charles, Holden C. Hutchinson, Melissa Schoenbaum, Marie R. Culhane, Carol J. Cardona.

**Methodology:** Amos Ssematimba, Sasidhar Malladi, Peter J. Bonney.

**Project administration:** Marie R. Culhane, Carol J. Cardona.

**Resources:** Carol J. Cardona.

**Software:** Sasidhar Malladi, Peter J. Bonney.

**Supervision:** Marie R. Culhane, Carol J. Cardona.

**Validation:** Amos Ssematimba, Sasidhar Malladi, Kaitlyn M. St. Charles, Rosemary Marusak.

**Visualization:** Amos Ssematimba, Sasidhar Malladi, Peter J. Bonney.

**Writing – original draft:** Amos Ssematimba.

**Writing – review & editing:** Sasidhar Malladi, Peter J. Bonney, Kaitlyn M. St. Charles, Holden C. Hutchinson, Melissa Schoenbaum, Rosemary Marusak, Marie R. Culhane, Carol J. Cardona.

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
