## [Decision Letter · Decision Letter 0]

9 Oct 2024

PONE-D-24-38916Estimating adequate contact rates and time of Highly Pathogenic Avian Influenza virus introduction into individual United States commercial poultry flocks during the 2022/24 epizooticPLOS ONE

Dear Dr. Ssematimba,

Thank you for submitting your manuscript to PLOS ONE. After careful consideration, we feel that it has merit but does not fully meet PLOS ONE’s publication criteria as it currently stands. Therefore, we invite you to submit a revised version of the manuscript that addresses the points raised during the review process.

We look forward to receiving your revised manuscript.

Kind regards,

Muhammad Ahmad

Academic Editor

PLOS ONE

2. Please note that PLOS ONE has spec6ific guidelines on code sharing for submissions in which author-generated code underpins the findings in the manuscript. In these cases, all author-generated code must be made available without restrictions upon publication of the work. Please review our guidelines at https://journals.plos.org/plosone/s/materials-and-software-sharing#loc-sharing-code and ensure that your code is shared in a way that follows best practice and facilitates reproducibility and reuse.

“This study was funded through a cooperative agreement between the Center for Epidemiology and Animal Health (CEAH) of the USDA, Animal and Plant Health Inspection Service (APHIS) Veterinary Services (VS) and the University of Minnesota (UMN) as USDA Award # AP22VSSP0000C051 (Quantitative Analysis and Support for Agricultural Animal Disease Emergency Preparedness and Response in the U.S.). The findings and conclusions in this document are those of the authors and should not be construed to represent any official USDA or U.S. Government determination or policy.”

Reviewers' comments:

Reviewer's Responses to Questions

**Comments to the Author**

1. Is the manuscript technically sound, and do the data support the conclusions?

Reviewer #1: Yes

Reviewer #2: Yes

2. Has the statistical analysis been performed appropriately and rigorously? 

Reviewer #1: Yes

Reviewer #2: Yes

3. Have the authors made all data underlying the findings in their manuscript fully available?

Reviewer #1: No

Reviewer #2: No

4. Is the manuscript presented in an intelligible fashion and written in standard English?

Reviewer #1: No

Reviewer #2: Yes

5. Review Comments to the Author

Reviewer #1: The manuscript presents an important analysis of the introduction time (TOI) and adequate contact rate (ACR) for HPAI outbreaks in layer, broiler, and turkey flocks in the US during the 2022/24 epizootic. The authors employ mathematical modeling techniques, including the Approximate Bayesian Computation (ABC) algorithm alongside a stochastic within-flock HPAI transmission model, to estimate the detection time of HPAI in flocks. While the study offers valuable insights, I have some suggestions and questions that need to be addressed to improve the manuscript (detailed in the attached file).

Reviewer #2: Title need to be short specific and simplified.

From line 18-23, 26-28, need to be modified.

Line 30 – related studies.

Line 33-34 The estimated parameters can also inform models used for surveillance design, risk analysis, and emergency preparedness.

Line 39 was preceded by detections in wild birds. What type of wild birds and the sentence need to be reformed. Citation not provided.

Line 40-41 writing error please check.

Line 140 please provide Citation.

Line 176-177 author mention about graph but didn’t mention any graph no in the manuscript. Citation required.

Line 180-184 author please clarify the sentence and need to be rewrite.

Line 204-205 citation error.

In methos analysis line 177-184 data does not match with the result showing in line no 207-210. Please clarify this.

In Discussion language very confusing please simplify the sentences to highlight your outcomes with the previously available database.

References are not properly cited in the main text.

6. PLOS authors have the option to publish the peer review history of their article (what does this mean?). If published, this will include your full peer review and any attached files.

Reviewer #1: **Yes: **Muhammad Haris Raza Farhan

Reviewer #2: **Yes: **Dr. Nisha beniwal

---

## [Author Response · Author response to Decision Letter 0]

16 Nov 2024

The files have been renamed as per the guidelines.

2. Please note that PLOS ONE has spec6ific guidelines on code sharing for submissions in which author-generated code underpins the findings in the manuscript. In these cases, all author-generated code must be made available without restrictions upon publication of the work. Please review our guidelines at https://journals.plos.org/plosone/s/materials-and-software-sharing#loc-sharing-code and ensure that your code is shared in a way that follows best practice and facilitates reproducibility and reuse.

The author-generated code will be readily without restrictions. 

“This study was funded through a cooperative agreement between the Center for Epidemiology and Animal Health (CEAH) of the USDA, Animal and Plant Health Inspection Service (APHIS) Veterinary Services (VS) and the University of Minnesota (UMN) as USDA Award # AP22VSSP0000C051 (Quantitative Analysis and Support for Agricultural Animal Disease Emergency Preparedness and Response in the U.S.). The findings and conclusions in this document are those of the authors and should not be construed to represent any official USDA or U.S. Government determination or policy.”

We have amended the financial disclosure to include the statement “…USDA co-authors assisted with data collection and preparation of the manuscript as part of work performed under the cooperative agreement. At the time of manuscript preparation, the cooperative agreement was not directly managed by any of the co-authors.…” in lines 393-396.

The data used as part of this study were collected by USDA as part of disease outbreak response activity purposes and are considered confidential business information. Anonymized data may be made available upon request, for statistical purposes only. Requests to access the datasets should be directed to HCH.

Review Comments to the Author

Reviewer #1: The manuscript presents an important analysis of the introduction time (TOI) and adequate contact rate (ACR) for HPAI outbreaks in layer, broiler, and turkey flocks in the US during the 2022/24 epizootic. The authors employ mathematical modeling techniques, including the Approximate Bayesian Computation (ABC) algorithm alongside a stochastic within-flock HPAI transmission model, to estimate the detection time of HPAI in flocks. While the study offers valuable insights, I have some suggestions and questions that need to be addressed to improve the manuscript (detailed in the attached file).

Estimating adequate contact rates and time of Highly Pathogenic Avian Influenza virus introduction into individual United States commercial poultry flocks during the 2022/24 epizootic (PONE-D-24-38916)

The manuscript presents an important analysis of the introduction time (TOI) and adequate contact rate (ACR) for HPAI outbreaks in layer, broiler, and turkey flocks in the US during the 2022/24 epizootic. The authors employ mathematical modeling techniques, including the Approximate Bayesian Computation (ABC) algorithm alongside a stochastic within-flock HPAI transmission model, to estimate the detection time of HPAI in flocks. While the study offers valuable insights, I have some suggestions and questions that need to be addressed to improve the manuscript.

Thank you for the compliments and the subsequent comments as they will help us improve the manuscript.

 General Comments:

The manuscript utilizes complex sentence structures that may impede reader comprehension. Simplifying these sentences will enhance clarity and improve understanding of the results. Additionally, several grammatical errors throughout the text should be rectified to elevate the manuscript's overall quality.

This has been addressed through several changes as highlighted in track changes version of the manuscript. Also, the effected changes are captured in the individual rebuttals below.

 Specific revisions:

Lines 38-40: Please revise for clarity and remove the extraneous bracket.

This has been addressed directly and also as part of the corrections suggested by the other reviewer in line 42-47 as follows: “…The 2022/24 H5N1 Highly Pathogenic Avian Influenza (HPAI) clade 2.3.4.4b outbreak in North American poultry and other species was preceded by detections in wild birds. The initial detection was from a deceased Great Black-backed Gull submitted to the Canadian Wildlife Health Cooperative in late 2021 (Giacinti et al., 2024); subsequently, the first detections in the United States occurred in January 2022 and were among wild ducks sampled for USDA active wild bird surveillance (USDA APHIS, 2024d). …”.

Line 61: Use the numeral '14' instead of spelling out 'fourteen.'

This has been changed accordingly in line 68.

Line 71: This citation appears unnecessary and should be removed.

The citation of (Center for Epidemiology and Animal Health, 2023) has been removed from line 78 as suggested.

Lines 74-77: The content is unclear and lacks coherence; please revise for better flow.

The content has been revised for improved clarity and coherence in lines 81-86 as “…To elucidate the possible discrepancies between the two outbreaks, the USDA performed case-control studies for commercial turkey (Patyk et al., 2023) and commercial table egg layer flocks (Green et al., 2023). Collectively from the two studies, the identified risk factors for HPAIV introduction included being in a disease control zone, increased sightings of wild waterfowl in the vicinity of the poultry premises, and offsite rendering of farm byproducts, among others….”

Lines 105-106: This sentence appears incomplete; please clarify.

This has been revised for improved clarity in lines 114-116 as “…The data used in this analysis included daily mortality records and diagnostic laboratory submission reports and test results….”

Line 114: Did you mean "Individual flock-based"? If so, please revise for accuracy.

The transmission model tracks every bird in the flock individually through the different disease stages that it undergoes. Such models are broadly referred to as “individual-based models”. However, in order to eliminate the ambiguity about the epidemiological unit of interest, we have adopted the term “individual bird-based model” and we have revised the sentence in lines 116-118 as “…A stochastic SEIR (i.e., susceptible, latently infected, infectious, and removed (dead)) individual bird-based HPAI transmission model was used to simulate the disease induced mortality and infection prevalence. …”

Lines 204-205: There seems to be an error in the citation; please check.

This error regrettably arose from the creation of the pdf version by the submission system. The original lines 233-235 read as follows “…For the reported index commercial poultry case of the current outbreak in the U.S., the observed and model predicted outcomes for mortality data are presented in Figure 1…”. 

Line 281: Remove the unnecessary comma.

The extra comma has been removed.

Line 321: Use either "However" or "Most importantly," but not both

In line 351, we have changed the phrasing to use “Notwithstanding” and the sentence now reads “…Notwithstanding, including these flocks in the analysis did not affect the trend observed in the species-specific comparison of outcomes….”

Authors Contribution section should be added

We have added the following to this effect in lines 403-408: 

“…Author Contributions

A.S. and S.M. performed the analyses and A.S. drafted the article. A.S., S.M., P.J.B., K.M.S., H.C.H., M.S., R.M., M.R.C., and C.J.C. conceived the analysis. H.C.H and M.S. acquired the data in coordination with field epidemiologists and response personnel. S.M., P.J.B., K.M.S., H.C.H., M.S., R.M., M.R.C., and C.J.C. provided substantive revisions to the article. All authors reviewed the manuscript and approved the final version submitted….” 

 Methods - Data: The 'Data' section does not provide sufficient detail regarding the data utilized in this study. It is essential to include the following information:

The total number of farms/flocks that were infected.

The distribution of farms by species, rearing system, and other relevant parameters.

The inclusion and exclusion criteria applied to the flock data.

Providing this information will enhance the transparency and reproducibility of your findings

This has been addressed through the following changes:

1) In data section, lines 114-128 as “…The data used in this analysis included daily mortality records and diagnostic laboratory submission reports and test results. Mortality records are provided to the USDA for indemnity purposes; laboratory documents for positive premises are recorded in USDA’s Emergency Management Response System. This data was deidentified and provided to the Secure Food Systems (SFS) Team at the University of Minnesota for analysis via a material transfer agreement. Flock data provided to the SFS team reflect a convenience sample of infected premises. Infected premises which were the first detection in a State and/or thought to be involved in cases of lateral transmission were selected for analysis to support ongoing field investigations. Furthermore, premises with inadequate data (e.g., mortality records provided in aggregate instead of daily, unknown sampling schemes) or that had abnormal baseline mortality patterns, likely due to underlying comorbidities (e.g., avian metapneumovirus), were excluded. Commercial premises infected between February 2022 and December 31, 2023 were eligible for inclusion. At this time, 452 commercial premises infected with HPAIV were confirmed detected. Detections included 39 commercial broiler premises, 55 commercial table egg layer premises, and 308 commercial turkey premises…”

2) In results’ section, lines 230-233 as “…The subset of premises analyzed reflect 18% (72 out of 402) of all infected broiler (9 out of 39), layer (18 out of 55), and turkey (45 out of 308) premises during the data period; split by epidemic phase, 24% (42 out of 174) of premises during end -July’22 and 13% (30 out of 228) of premises post-July’22 were analyzed….”

 Results: The results section could benefit from clearer organization. Consider dividing the results into specific sub-sections or sub-headings to facilitate better understanding. 

We have made the following changes to this effect: We have changed the very first paragraph in results in lines to be read “…The subset of premises analyzed reflect 18% (72 out of 402) of all infected broiler (9 out of 39), layer (18 out of 55), and turkey (45 out of 308) premises during the data period; split by epidemic phase, 24% (42 out of 174) of premises during end -July’22 and 13% (30 out of 228) of premises post-July’22 were analyzed…”. Then the second paragraph introduces the example graph on the comparison between the model output and field derived mortality data, the next paragraphs introduce results in Table 2 which presents an overall summary of the number of test days and the results on the estimated MTFPS, within-flock ACR, and R_0. After presenting the overall results, we then zoom in on the species-specific results followed by the epidemic-phase specific results. Lastly we present species-specific results grouped by the ranking of the adequate contact rate into slow, moderate and fast spread.

 Specific Questions: 

 Did your analyses include any back-calculation methods? If so, please elaborate on how these were integrated into your study.

The study did not use a back-calculation method. It used a “forward simulation approach” in which the model predicted outcomes for a given set of parameters drawn from the given prior distributions were compared with the observed field data as described in the methods. Using a distance measure, the set of parameter values meeting the inclusion criteria were selected and summarized as outputs of interest. Note that this forward simulation approach enables considering the variability in disease durations among individual birds more directly and is beneficial to jointly estimate the adequate contact rate and the time of virus introduction.

 When comparing your results to other studies, particularly the Dutch 2003 epidemic, how do you account for differences in data, including strain type, geographical region, and variations in self-reporting or testing days?

We address this by emphasizing caution when attempting to compare results involving different virus strains, different farm management practices as is the case between Europe and the U.S. We only mention the highlighted studies as being “closely related” e.g. in terms of methods and/or the data used as well as species and generic HPAI disease. To this effect, we stated in lines 308-315 that “…However, comparing findings from studies involving different virus strains should be done with caution. Note that HPAI virus strain-specific characteristics (Spickler et al., 2008), production type, and flock management practices may influence within flock disease spread dynamics. As examples of closely related studies using field HPAI outbreak mortality data in commercial turkey flocks to estimate disease parameters, an ACR of 3.2 (95% CI: 2.3, 4.3) per day and R_0 of 12.8 (95% CI: 9.2, 17.2) were estimated for the 2014/15 U.S. H5N2 epidemic (Ssematimba et al., 2019) and, an ACR of 3.37 (95% CI: 0.97, 11.74) per day was estimated for the Dutch 2003 H7N7 epidemic (Bos et al., 2009)….”

Addressing these points will strengthen the manuscript and provide clearer insights into your research. Thank You!

Indeed these comments have been very well received and adjustments to the manuscript have been made. We greatly appreciate the efforts dispensed. 

Reviewer #2: Title need to be short specific and simplified.

The title has be shortened to: “Estimating the time of Highly Pathogenic Avian Inf

---

## [Editor Report · Decision Letter 1]

22 Nov 2024

Estimating the time of Highly Pathogenic Avian Influenza virus introduction into United States poultry flocks during the 2022/24 epizootic

PONE-D-24-38916R1

Dear Dr. Ssematimba,

We’re pleased to inform you that your manuscript has been judged scientifically suitable for publication and will be formally accepted for publication once it meets all outstanding technical requirements.

Kind regards,

Muhammad Ahmad

Academic Editor

PLOS ONE
---

## [Editor Report · Acceptance letter]

4 Dec 2024

PONE-D-24-38916R1 

PLOS ONE

Dear Dr. Ssematimba, 

I'm pleased to inform you that your manuscript has been deemed suitable for publication in PLOS ONE. Congratulations! Your manuscript is now being handed over to our production team.

Kind regards, 

on behalf of

Mr. Muhammad Ahmad 

Academic Editor

PLOS ONE